# Cognitive Effects of Reducing First-Generation Antipsychotic Dose Compared to Switching to Ziprasidone in Long-Stay Patients with Schizophrenia

**DOI:** 10.3390/jcm13072112

**Published:** 2024-04-04

**Authors:** Jan P. A. M. Bogers, Jasper A. Blömer, Lieuwe de Haan

**Affiliations:** 1High Care Clinics and Rivierduinen Academy, Mental Health Services Rivierduinen, P.O. Box 405, 2300 AK Leiden, The Netherlands; 2High Care Clinics, MHS Rivierduinen, Leiden, and PsyQ and Brijder Addiction Care, 2034 MA Haarlem, The Netherlands; j.blomer@psyq.nl; 3Department of Psychiatry, Amsterdam University Medical Center, 1105 AZ Amsterdam, The Netherlands; l.dehaan@amsterdamumc.nl

**Keywords:** long-stay schizophrenia, cognition, cognitive function, antipsychotics, ziprasidone, dose reduction, neuropsychology, recovery

## Abstract

**Background:** Cognitive impairment is a core symptom of schizophrenia and is associated with functional outcomes. Improving cognitive function is an important treatment goal. Studies have reported beneficial cognitive effects of the second-generation antipsychotic (SGA) ziprasidone. Reducing the dose of first-generation antipsychotics (FGA) might also improve cognitive function. This study compared the cognitive effects in long-stay patients who were randomized to groups who underwent FGA dose reduction or switched to ziprasidone. **Methods**: High-dose FGA was reduced to an equivalent of 5 mg of haloperidol in 10 patients (FGA-DR-condition), and 13 patients switched to ziprasidone 80 mg b.i.d. (ZIPRA condition). Five domains of cognitive function were assessed before dose reduction or switching (T0) and after 1 year (T1). This study was approved by the ethics committee of the Open Ankh (CCMO number 338) and registered at the Netherlands Trial Register (code 5864). **Results**: Non-significant deterioration was seen in all cognitive domains studied in the FGA-DR condition, whereas there was a non-significant improvement in all cognitive domains in the ZIPRA condition. The most robust difference between conditions, in favor of ziprasidone, was in executive function. **Conclusions**: In patients with severe chronic schizophrenia, ziprasidone had a non-significant and very modest beneficial effect on cognitive function compared with FGA dose reduction. Larger trials are needed to further investigate this effect.

## 1. Introduction

One of the most disabling symptoms of schizophrenia is the impairment of cognitive function [1,2,3]. This should receive full attention in treatment since cognitive function is associated with functional outcomes [4]. Improving cognitive function is an important treatment goal.

Although cognitive deficits in general are common in schizophrenia, impairments in processing speed, working memory, verbal learning, and executive function are frequently reported [5,6,7,8].

The pharmacological treatment of schizophrenia mostly consists of dopamine-receptor antagonists, which mitigate positive symptoms, and to a lesser extent, negative symptoms, but most studies suggest that neither first-generation antipsychotics (FGAs) nor second-generation antipsychotics (SGAs) substantially improve cognitive deficits [9,10,11]. This is confirmed by the results of large real-world trials where modest improvements were found without a significant benefit of SGA over FGA [12,13].

However, specific SGAs have been found to improve some cognitive domains, although they failed to demonstrate substantial overall efficacy for relieving cognitive dysfunction in schizophrenia [11]. There is a considerable variety in the reported effects of individual antipsychotics [14,15]. Some studies have reported that the SGAs quetiapine, risperidone, and olanzapine have beneficial effects [16,17]. Reviews have reported that ziprasidone has beneficial effects on cognitive symptoms [14,18,19], as have trials involving patients with a short duration of illness [20,21,22] and long-stay patients [23,24,25,26,27]. In patients treated with ziprasidone, compared with patients who underwent treatment with other antipsychotics, improvements in several cognitive domains were reported, such as executive function [18,19,23,25], processing speed [14,18,19,25], attention and vigilance [14,18,23], working memory [14,18,19], and verbal learning [19,23]. Some studies have reported that cognitive improvements occur separately from improvements in other symptoms, suggesting that ziprasidone improves cognitive function independently of symptomatic improvement [20,23,24,25].

Since cognitive impairment is also associated with the dose of antipsychotic used [28,29], lowering the dose might improve cognitive deficits. However, it is not known how low the dose has to be to improve cognitive symptoms. A daily dose of 5 mg of haloperidol or an equivalent dose of another antipsychotic (5 mg HE) is considered an adequate dose to treat psychotic symptoms in patients with multi-episode or chronic schizophrenia based on trials, studies of dopamine D_2_ receptor occupancy, and expert opinion [30,31,32,33].

To the best of our knowledge, there have been no studies comparing the cognitive effects of reducing the dose of FGAs to 5 mg of HE per day or switching to ziprasidone in patients currently being treated with high doses of FGAs. To this end, we analyzed data on cognitive function that were collected in a trial primarily focusing on negative symptoms, and we compared a dose reduction strategy to achieve an FGA dose of 5 mg of HE per day with switching to an equivalent dose of ziprasidone. We hypothesized that (1) reducing the dose of FGAs would improve cognitive function and (2) switching to ziprasidone would have a superior effect on cognitive function than FGA dose reduction.

## 2. Materials and Methods

### 2.1. Study Design

In this preliminary study, patients were randomized to one of two treatments. Patients in the dose reduction group (*n* = 10) received their current FGAs, but their doses were slowly reduced to a target of 5 mg of HE/day (FGA-DR-condition). Patients in the switch group (*n* = 13) received ziprasidone at a dose of 160 mg/day, equivalent to 5 mg of haloperidol, given as 80 mg b.i.d. (ZIPRA-condition). This study was part of a clinical research project investigating FGA dose reduction or switching to ziprasidone in long-stay patients with severe chronic schizophrenia treated with high doses of FGAs. The original study was a double-blind randomized study with a 1-year follow-up, with negative symptoms as the primary outcome. The present study focused on neurocognitive function. Study methods have previously been described in detail [34]. Figure 1 shows a timetable of this study.

### 2.2. Study Participants

The participants of this clinical research project were patients who had severe disabling symptoms, such as aggressive behavior and/or deficit states. Most patients were involuntarily hospitalized in long-stay clinics. These patients had been prescribed various high-dose FGAs for many years. The overall aim of this study was to evaluate whether dose reduction or treatment with ziprasidone would improve clinical outcomes. None of the patients were using clozapine at baseline or had used clozapine in the past, even though they experienced refractory positive and negative symptoms for years. All participants had a DSM-IV diagnosis of schizophrenia or schizoaffective disorder based on a Structured Clinical Interview for DSM disorders (SCID). According to the inclusion criteria, they were older than 18 years, had persistent symptoms for at least 2 years, were on stable medication (no change 3 months prior to inclusion in the study), and were able to comply with the study protocol. All patients gave their consent to use oral capsules of medication or placebo. None had diseases that could pose a medical risk, such as a QTc interval of more than 500 ms, which increases the risk of ziprasidone-associated arrhythmias.

Throughout the study, study medication was given under double-blind conditions. The medication/placebo capsules were prepared by the Central Hospital Pharmacy in The Hague. They were identical in appearance, taste, and smell.

### 2.3. Drug Regimen and Assessments

This study had three phases (Figure 1). At baseline, patients were assessed before their medications were changed. In the next phase (the dose-adjustment or switch phase), their medications were changed. In the FGA-DR condition, the dose was gradually reduced to 5 mg of HE/day based on each patient’s starting dose, but was maximally reduced over 24 weeks for baseline FGA doses higher than 30 mg of HE/day (according to a dose reduction table for each individual patient based on the starting dose). It was expected that this slow dose reduction scheme would prevent withdrawal psychosis. In the ZIPRA condition, the current FGA treatment was changed in two steps: 2.5 mg of HE was replaced with ziprasidone at a dose of 80 mg/day, and after 6 weeks, a further 2.5 mg of HE was replaced with ziprasidone at a dose of 80 mg/day (total daily dose of 80 mg b.i.d.). The remaining FGA dose was gradually withdrawn (until 0 mg was reached) as described for the FGA-DR condition. The final treatment and observation phase lasted 1 year.

Patients continued to take non-study medication used at baseline that remained unchanged throughout the study, with the exception of anticholinergic drugs, which were gradually withdrawn during the dose-adjustment and switch phase, but they were reintroduced if the patient experienced extrapyramidal side effects.

### 2.4. Clinical Assessments

Cognitive function was assessed at baseline before dose reduction or switch (T0). Endpoint assessment was performed after 1 year (T1). The cognitive assessment battery evaluated cognitive function in five domains: (1) speed of processing information, (2) attention and vigilance, (3) working memory, (4) verbal learning, and (5) executive function (Table 1).

A brief description of the tests used to assess cognitive function can be found in Appendix A.

All testing was performed by the same psychologist who was blinded to the treatment condition.

None of the patients were rapid metabolizers based on P450 CYP2D6 enzyme measurements. Therefore, rapid metabolism did not explain the need for high antipsychotic doses, nor was it a contraindication for dose reduction.

### 2.5. Analysis

All analyses were carried out using IBM SPSS Statistics, Version 20.0. Baseline demographic and clinical differences were tested using the Fisher exact test (sex) and the Mann–Whitney U-test (age, duration of admission, antipsychotic use, and clinical severity score). The condition differences and the differences in the effects of each treatment were assessed by applying crude neuropsychological test scores. Because of the small number of patients, non-parametric testing was applied. Non-parametric testing uses median values and ranks outcomes, which can lead to medium or large effect sizes, while the median values (hardly) differ. Changes in neuropsychological scores between T0 and T1 for all patients, patients in the FGA condition, and patients in the ZIPRA condition were analyzed using the Wilcoxon signed rank test. Differences between conditions were analyzed with the Mann–Whitney U-test, using differences between crude scores (ΔT0-T1). Also, for both conditions, effect sizes (r) between time points T0 and T1 and between conditions for all cognitive tests were calculated as follows: r=Z/√N. The effect size is a quantitative measure of the magnitude of the experimental effect and therefore indicates the magnitude of differences between times of testing or between conditions. The Pearson r coefficient is considered to be low if the value of r varies by about 0.10, medium if it varies by about 0.30, and large if it varies by more than 0.50 [35]. It is important to note that the effect sizes are reported as negative numbers in the applied statistical program.

### 2.6. Ethical Standard

The Institutional Review Board of Rivierduinen and the Ethics Committee of the Open Ankh (CCMO number 338) approved this study, which was prospectively registered at the Netherlands Trial Register (code 5864). After receiving a detailed explanation of the trial, the participants or their legal representatives gave their written informed consent.

## 3. Results

### 3.1. Patients

In the original trial, 48 patients were randomized. During the dose-adjustment or switch phase and the treatment and observation phase, 10 patients were withdrawn from the study because of treatment failure, i.e., a prolonged or repeated relapse [34]. In addition, data were missing for six patients at T0 or T1 and for nine patients at T0 and T1. This high rate of missing outcome measures is explained by the characteristics of the included patients, who were long-stay patients with severe, disabling symptoms and an impaired ability to perform cognitive tests, and they were inclined to refuse cognitive testing. Sufficient data were collected from 23 patients: 10 in the FGA-DR condition and 13 in the ZIPRA condition.

### 3.2. Patient Characteristics

All patients had severe symptoms (mean baseline PANSS score of 104 in the FGA-DR condition and 97 in the ZIPRA condition) and had undergone involuntary in-hospital treatment for a large number of years. The patients’ demographic data and characteristics are shown in Table 2. There were no significant differences in the baseline characteristics of the two patient groups. The mean duration of the treatment phase from 4 weeks after the dose reduction or switch phase to T1 was 57.1 weeks (SD 9.9), and this was not different between conditions (Z = −0.92; r = 0.19; *p* = 0.93).

### 3.3. Does Reducing the Dose of FGAs Improve Cognitive Function?

There was a non-significant deterioration in all cognitive domains in patients receiving FGA-DR. The effect size was large in three domains (speed of processing information, verbal learning, and executive function; r > 0.50) and medium in two domains (attention and vigilance and working memory) (Table 3).

### 3.4. Does Switching to Ziprasidone Have a Superior Effect on Cognitive Function Than FGA Dose Reduction?

Differences in the change in cognitive function between T0 and T1 between the FGA-DR condition and the ZIPRA condition were observed for executive function (verbal fluency letter A), in favor of ziprasidone, with a large effect size (r = 0.54) (Table 4). Similar patterns were found for other verbal fluency assessments and in the domain of attention and vigilance. These between-condition differences were not significant despite having a medium or large effect size (r = 0.33 and 0.59, respectively, 0.44), and they were associated with a non-significant improvement in all cognitive domains in the ZIPRA condition (Table 3). The most robust between-condition difference was seen in the domain of executive function and was in favor of ziprasidone (Table 4).

## 4. Discussion

We included patients who were severely ill and highly dosed with long-stay schizophrenia and impaired cognitive function in all domains. We found no significant changes in cognitive function after 1 year of treatment with 5 mg of HE FGA or ziprasidone. Although non-significant, there was a deterioration in most cognitive function domains in patients in the FGA-DR condition, whereas patients taking ziprasidone showed improvement, although not significant, with medium and large effects sizes, mainly in the domain of executive function.

Thus, reducing the dose of FGA did not improve cognitive function but, in contrast, led to a non-significant deterioration in most domains. While both treatments can be considered dose reduction conditions, the improvement seen with ziprasidone in most domains, more explicitly in executive function, if replicated in a larger trial, can be attributed to that drug and not to dose reduction.

It has been claimed that executive dysfunction constitutes the most specific set of neuropsychological symptoms in schizophrenia [6,7,8]. Improvements in cognitive function are associated with the recovery of patients. For instance, a recent study found that personal recovery was positively related to self-reported executive function [36]. While improving patients’ functional outcomes remains an important treatment goal, attention should be paid to improving their cognitive function.

The between-condition differences in favor of ziprasidone, albeit non-significant, provide preliminary support for carrying out a larger trial to test the second hypothesis, namely, that ziprasidone has a superior effect on cognitive function than reducing the dose of FGAs. At the same time, because most cognitive assessments did not differ significantly or had small or medium effect sizes, our findings suggest that differences between the cognitive effects of reducing the dose of FGAs and ziprasidone might be of limited clinical relevance. The hint of a favorable effect of ziprasidone is in line with the findings of previous studies, although these studies mainly found improvements in working memory and verbal learning [14,20], which are domains in which we found improvements with a medium effect size. Because the sample size was small, our power to detect significant differences was low. However, given the substantial effect sizes, further evaluation is warranted.

Although the original study was a randomized trial, this analysis of cognitive function data should be considered as a preliminary study in a specific, involuntarily admitted, long-stay, difficult-to-treat (and investigate) patient population with chronic psychosis. Only the most cooperative and stable patients were able to comply with the research protocol of testing cognitive function, while others were not able or not willing to participate, which resulted in a high dropout rate for the cognitive function tests. This may reduce the comparability of the two groups of patients, which could generate a risk of attrition bias, although the baseline characteristics of the two groups did not differ significantly.

This study had a number of strong points, with the first being the study population. All of the patients had stable schizophrenia, confirmed with an SCID interview, but were very ill and had been involuntarily admitted to the hospital for more than 15 years. This is a neglected population in research with a high illness burden. Second, we reduced the antipsychotic dose to 5 mg of HE per day, which is a dose considered by most experts to be adequate, using a slow and personalized dose reduction scheme to prevent relapse or withdrawal symptoms [37]. Third, we used a relatively high dose of ziprasidone that was equivalent to 5 mg of HE. This enabled us to make group comparisons because the patients in both conditions had been treated with equivalent doses. Fourth, blinding was retained throughout the study. Fifth, by assessing CYP2D6 activity, we could exclude rapid metabolism as an explanation for the high doses prescribed to patients at baseline, and this confirms that we compared equivalent doses between conditions. Sixth, since all patients were admitted, we were able to control treatment compliance. Lastly, the time between the baseline and the end measurement was about 1 year, which should have been sufficiently long for cognitive function to have stabilized on a reduced dose of FGAs or ziprasidone. However, it should be noted that the literature is not unanimous concerning the necessary time between medication changes and cognitive testing, varying between 4 weeks and longer [20].

However, our study had some limitations. There were relatively few patients, which limited our power to detect significant differences. Second, the inclusion of a specific patient group in a specific treatment setting limits the generalizability of the findings. In this respect, the patients were relatively old (the mean ages in both groups were 50 and 53 years, respectively). Third, given the relatively high ages of the participants, it cannot be ruled out that cognitive impairment, as a result of neurological disorders, was a confounding factor. Fourth, while the dose of FGAs was reduced in both conditions, we cannot be sure that the changes in cognitive function were mainly driven by the dose reduction or switch because we did not include a control group that was kept on high-dose FGA. Fifth, a dose reduction from approximately 15 mg of HE to 5 mg of HE per day is a substantial reduction, but it might have been insufficient to produce large improvements. Sixth, we did not control for anticholinergic use, which is a possible confounding factor. While we did not change the use of other medications, it was possible to reduce the dose of anticholinergic drugs if they were no longer considered necessary because extrapyramidal symptoms diminish with lower antipsychotic doses. Because anticholinergic agents impair cognitive function, this might have influenced the outcomes, but this was unlikely because the use of anticholinergic agents was decreased in both treatment groups. Seventh, we cannot exclude that the learning effects influenced the outcomes (performance bias). Other studies have found that cognitive function improves in people who take antipsychotic agents, with the effect sizes being comparable to those of the learning effects [12,38]. Lastly, we did not register the educational levels of our patients, so we cannot control or compare the test results with those of norm groups.

## 5. Conclusions

In this small prospective study of involuntarily treated patients with longstanding severe chronic schizophrenia, reducing the dose of FGAs did not have a beneficial effect on cognitive function, whereas treatment with ziprasidone seemed to improve executive function. A larger trial is urgently needed to investigate the effects of ziprasidone on cognitive function in this specific patient group before firm conclusions about the potential beneficial effect of ziprasidone can be drawn.

## Figures and Tables

**Figure 1 jcm-13-02112-f001:**
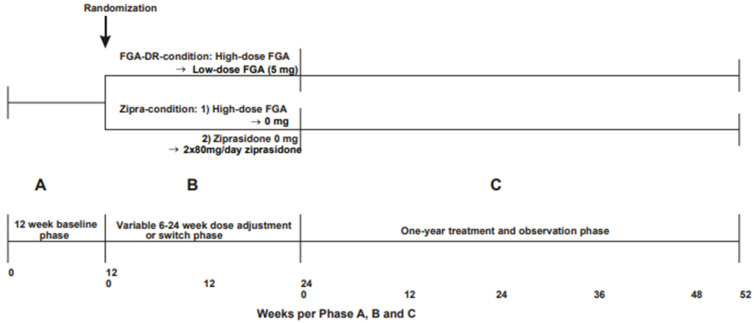
Timetable of trial. FGA, first-generation antipsychotic; DR, dose reduction.

**Table 1 jcm-13-02112-t001:** Cognitive domains tested, applied tests, and number of participating patients.

Cognitive Domain:	Test:	Number of Participants		
		Total Group	FGA-DR Condition	ZIPRA Condition
1. Speed of processing information	Symbol digit modalities test Stroop color word test, card 1 Trail making test part A	18 19 20	7 9 9	11 10 11
2. Attention and vigilance	Continuous performance test	11	5	6
3. Working memory	Digits forward Digits backward	23	10	13
4. Verbal learning	15 words test (Rey auditory verbal learning test)	21	9	12
5. Executive functions	Verbal fluency tests Stroop interference (cards 2 and 3) Trail making test part B	21 19 20	9 9 9	12 10 11

FGA, first-generation antipsychotic; ZIPRA, ziprasidone; DR, dose reduction.

**Table 2 jcm-13-02112-t002:** Baseline characteristics of study population.

	FGA-DR Condition (*n* = 10)	ZIPRA Condition (*n* = 13)
Male in numbers (%)	7 (70)	6 (46)
Mean age in years (SD)	50.00 (17.51)	52.62 (14.09)
Mean time (years) in the hospital (SD)	17.38 (10.62)	18.67 (12.94)
Mean time (years) of antipsychotic use	17.38 (10.62)	19.33 (12.63)
Dose equivalents in mg/day of haloperidol before dose reduction/switch (SD)	16.95 (15.98)	13.37 (9.56)
Mean PANSS total score (SD)	103.6 (10.68)	96.54 (7.21)

SD, standard deviation; FGA, first-generation antipsychotic; ZIPRA, ziprasidone; DR, dose reduction; PANSS, positive and negative syndrome scale.

**Table 3 jcm-13-02112-t003:** Test outcomes for total group of patients, patients in FGA (first-generation antipsychotics)-DR (dose reduction) condition, and patients in ziprasidone condition. Median values are shown for T0 and T1, and effect sizes (r) are shown for difference between T0 and T1.

	Total Group of Patients					FGA-DR Condition					Ziprasidone Condition				
	Median		∆T0-T1			Median		∆T0-T1			Median		∆T0-T1		
Domains	T0	T1	Z	*r*	*p*	T0	T1	Z	*r*	*p*	T0	T1	Z	*r*	*p*
1. Speed of processing information															
Symbol Digits MT	18.0	12.0	−2.25	−**0.53**	0.24	18.0	12.0	−1.36	−**0.51**	0.17	18.0	12.0	−1.78	−**0.53**	0.08
Stroop Chart 1	80.0	80.0	−0.47	−0.11	0.64	80.0	80.0	−0.71	−0.24	0.48	88.5	76.0	−0.28	−0.08	0.78
Trail Making Test A	94.0	96.5	−0.73	−0.16	0.47	94.0	114.0	−0.53	−0.18	0.59	94.0	88.0	−0.71	−0.21	0.48
2. Attention and vigilance															
Continuous Performance	698.0	647.0	−0.80	−0.24	0.42	605.0	727.0	−0.67	−0.29	0.50	826.5	622.5	−0.94	−0.38	0.35
3. Working memory															
Digits Forward	5.0	5.0	−1.03	−0.21	0.30	5.0	4.5	−0.36	−0.11	0.72	5.0	5.0	−1.29	−0.36	0.20
Digits Backward	3.0	3.0	−1.44	−0.30	0.15	3.0	3.0	−1.61	−0.50	0.11	4.0	3.0	−0.52	−0.14	0.61
4. Verbal learning															
15 Words Test Total	20.0	21.0	−1.15	−0.24	0.25	19.0	16.0	−0.36	−0.12	0.72	21.0	22.0	−1.10	−0.31	0.27
15 Words Test Recall	3.0	4.0	−1.21	−0.26	0.23	3.0	3.0	−0.11	−0.03	0.92	2.5	4.0	−1.50	−0.43	0.14
15 Words Test Recognition	13.0	12.0	−1.62	−0.35	0.11	13.0	11.0	−2.13	−**0.71**	0.33	12.5	12.5	−0.36	−0.11	0.72
5. Executive function															
Verbal fluency Letter N	5.0	4.0	−0.65	−0.14	0.52	3.0	3.0	−0.21	−0.07	0.83	5.0	5.0	−1.12	−0.32	0.27
Verbal fluency Letter A	3.0	3.0	−0.69	−0.15	0.49	3.0	2.0	−2.16	−**0.72**	0.31	4.0	5.5	−1.29	−0.37	0.20
Verbal fluency Animals	10.0	9.0	−1.14	−0.25	0.26	10.0	8.0	−1.56	−**0.52**	0.12	8.5	12.5	−2.50	−**0.72**	0.01 *
Verbal fluency Occupations	5.0	5.0	−0.88	−0.19	0.38	4.0	5.0	−0.68	−0.23	0.50	5.5	6.0	−1.58	−0.46	0.12
Stroop Interference Score	80.0	83.0	−0.21	−0.05	0.83	96.0	71.0	−0.42	−0.14	0.68	75.0	80.0	−0.14	−0.04	0.89
Trail Making Test B	346.0	287.0	−1.23	−0.28	0.22	342.0	269.0	−0.42	−0.14	0.68	351.0	305.0	−1.42	−0.43	0.16

Symbol Digits MT, Symbol Digits Modalities Test. Stroop interference = card 3 minus card 2. Effect sizes are considered to be low if r varies around 0.10, medium if it varies around 0.30, and large if >0.50. For all tests, higher scores mean better performances, except for Stroop charts, trail making tests, and continuous performance test. * = significant difference (*p* < 0.05), not significant after correction for multiple testing. Large effect sizes are in bold.

**Table 4 jcm-13-02112-t004:** Median differences between treatment with first-generation antipsychotics (FGA) and ziprasidone, and effect sizes (r) for difference between T0 and T1. Symbol Digits MT, Symbol Digits Modalities Test. Stroop interference = card 3 minus card 2. * = significant difference (*p* < 0.05), not significant after correction for multiple testing. Large effect sizes are in bold.

	∆T0-T1				
	Median Difference				
	FGA	Ziprasidone	*Z*	*r*	*p*
1. Speed of information processing					
Symbol Digits MT	−2.00	−5.00	−0.18	−0.04	0.86
Stroop Chart 1	−6.00	−5.00	−0.96	−0.22	0.92
Trail Making Test A	−12.00	−13.00	−0.49	−0.11	0.62
2. Attention and vigilance					
Continuous Performance	17.00	−207.00	−1.46	−0.44	0.14
3. Working memory					
Digits Forward	−5.00	0.00	−0.60	−0.12	0.55
Digits Backward	−2.00	−1.00	−1.01	−0.21	0.31
4. Verbal learning					
15 Words Test Total	3.00	7.50	−0.82	−0.18	0.41
15 Words Test Recall	0.00	0.50	−0.83	−0.18	0.41
15 Words Test Recognition	−2.00	0.00	−1.22	−0.26	0.22
5. Executive function					
Verbal Fluency Letter N	0.00	1.00	−0.54	−0.12	0.59
Verbal Fluency Letter A	−2.00	0.00	−2.50	**−0.54**	0.01 *
Verbal Fluency Animals	−1.00	2.50	−1.55	−0.33	0.12
Verbal Fluency Occupations	0.00	1.00	−2.72	**−0.59**	0.07
Stroop Interference Score	1.00	5.00	−0.48	−0.11	0.63
Trail Making Test B	−28.00	−71.00	−0.61	−0.14	0.54

## Data Availability

The data that support the finding of this study are available from the corresponding author, J.P.A.M.B., upon reasonable request.

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
