# Peer review of "Cognitive Effects of Reducing First-Generation Antipsychotic Dose Compared to Switching to Ziprasidone in Long-Stay Patients with Schizophrenia"

_jcm, 2024, doi:10.3390/jcm13072112_

Round 1

Reviewer 1 Report

Comments and Suggestions for Authors

Dear Authors, 

I have read with interest your research “Cognitive effects of dose-reduction of first-generation antipsychotics compared to switch to ziprasidone in long-stay schizophrenia patients”. This is an insightful and well conducted double-blind trial, in which non-significant deterioration of cognitive function was detected in the former group, whereas non-significant improvement in executive function in was found in the latter. Despite several limitations, which were opportunely declared in the Discussion, the findings could foster the design of a larger trial in order to further increase the generalizability of the results. 

Below, I point out a few suggestions and comments: 

Methods

1.     A figure showing the timetable of the three phases of the study could be helpful. 

2.     A brief explanation of the assessment tools used in this research could be helpful as well.

3.     Table 1, please remove the bold font in the item 1.

Discussion

4.     Given the relatively high age of participant, a mention about cognitive impairment due to dementia/neurological issues should be reported among limitations and biases. 

In conclusion, please sort the page numbering. 

Reviewer 2 Report

Comments and Suggestions for Authors

  • Clarify the rationale for the study and its significance in the introduction.
  • Provide more detailed descriptions of the methodology and participant characteristics.
  • Clearly define the cognitive domains assessed and the assessment tools used.
  • Present the results with more statistical analysis and interpretation.
  • Discuss potential confounding factors that may have influenced the results.
  • Consider discussing the clinical implications of the findings in more depth.
  • Provide clearer recommendations for future research, including the need for larger trials.
  • Improve the organization and flow of the manuscript for better readability.
  • Ensure consistency in terminology and definitions throughout the manuscript.
  • Proofread the manuscript for grammar, syntax, and formatting errors.

Comments on the Quality of English Language

Minor editing of English language required

Reviewer 3 Report

Comments and Suggestions for Authors

Brief summary: This interesting manuscript focuses on the comparison of using lower dose of first-gen antipsychotics to a group of patients who switched their regiment to ziprasione on the impact on cognitive domains. They explored five domains of cognitive function during the course of a year. Non-significant deterioration was observed in all the cognitive domains in the first group whereas the second group (ziprasidone) had non-significant improvement in all the domains studied. While the results presented does not highlight significant results, the authors highlight that larger trials could investigate this difference. 

Please see my comments below.

Introduction:

1. The introduction could benefit from further development on cognitive functions : what are the important cognitive domains established and studied in schizophrenia and what were the main results. This could be inserted prior to the second paragraph.

2. Considering the main focus of the study, it would be of interest to the readership if the authors further develop prior findings on ziprasidone (lines 56-60).

3. While this might be obvious for clinicians, a sentence or two on the reason why cognitive impairement is an important topic in the field of schizophrenia could further enhance the necessity for the authors' study. 

Overall the introduction is very clear, well presented and is easy to read.

Materials and Methods:

1. How was the sample size selected  (10 vs 13)? How were they recruited (I understand they were on long-stay units, but how were they approached and in which context). 

2. Why was the DSM-IV used instead of the current version (DSM-5)? It is unclear to me when were the patients recruited. I understand it was part of a prior study, but reading the original study (ref 30), this information is not readily accessible to the readership.

3. Patients who were on other medication for physical conditions or other psychiatric conditions interacting with CYP 2D6: how was this handled in order to not have an impact on your results? (apart from what was mentionned lines 146-149).

Results:

1. The results are very well presented. I have no comment on this section.

Discussion:

1. Findings are clearly discussed with the relevant literature and the limitations are well outlined.

This manuscript was really pleasant to read and easy to follow which is one of its great strength.

Minor comment:

The iThenticate reports matches 12% with the following publication made by the same author:

https://www.sciencedirect.com/science/article/pii/S0924977X18301652?via%3Dihub
